# Different Involvement of Vimentin during Invasion by *Listeria monocytogenes* at the Blood–Brain and the Blood–Cerebrospinal Fluid Barriers In Vitro

**DOI:** 10.3390/ijms232112908

**Published:** 2022-10-26

**Authors:** Franjo Banovic, Sandrin Schulze, Mobarak Abu Mraheil, Torsten Hain, Trinad Chakraborty, Véronique Orian-Rousseau, Selina Moroniak, Christel Weiss, Hiroshi Ishikawa, Horst Schroten, Rüdiger Adam, Christian Schwerk

**Affiliations:** 1Pediatric Infectious Diseases, Department of Pediatrics, Medical Faculty Mannheim, Heidelberg University, 68167 Mannheim, Germany; 2Institute of Medical Microbiology, German Center for Infection Research (DZIF), Partner Site Giessen-Marburg-Langen, Justus-Liebig University Giessen, 35392 Giessen, Germany; 3Institute of Biological and Chemical Systems-Functional Molecular Systems (IBCS-FMS), Karlsruhe Institute of Technology (KIT), 76344 Eggenstein-Leopoldshafen, Germany; 4Department of Medical Statistics and Biomathematics, Medical Faculty Mannheim, Heidelberg University, 68167 Mannheim, Germany; 5Laboratory of Clinical Regenerative Medicine, Department of Neurosurgery, Faculty of Medicine, University of Tsukuba, Tsukuba 305-8575, Japan

**Keywords:** blood–cerebrospinal fluid barrier, choroid plexus, internalin F, *Listeria monocytogenes*

## Abstract

The human central nervous system (CNS) is separated from the blood by distinct cellular barriers, including the blood–brain barrier (BBB) and the blood–cerebrospinal fluid (CFS) barrier (BCSFB). Whereas at the center of the BBB are the endothelial cells of the brain capillaries, the BCSFB is formed by the epithelium of the choroid plexus. Invasion of cells of either the BBB or the BCSFB is a potential first step during CNS entry by the Gram-positive bacterium *Listeria monocytogenes* (*Lm*). *Lm* possesses several virulence factors mediating host cell entry, such as the internalin protein family—including internalin (InlA), which binds E-cadherin (Ecad) on the surface of target cells, and internalin B (InlB)—interacting with the host cell receptor tyrosine kinase Met. A further family member is internalin (InlF), which targets the intermediate filament protein vimentin. Whereas InlF has been shown to play a role during brain invasion at the BBB, its function during infection at the BCSFB is not known. We use human brain microvascular endothelial cells (HBMEC) and human choroid plexus epithelial papilloma (HIBCPP) cells to investigate the roles of InlF and vimentin during CNS invasion by *Lm*. Whereas HBMEC present intracellular and surface vimentin (besides Met), HIBCPP cells do not express vimentin (except Met and Ecad). Treatment with the surface vimentin modulator withaferin A (WitA) inhibited invasion of *Lm* into HBMEC, but not HIBCPP cells. Invasion of *Lm* into HBMEC and HIBCPP cells is, however, independent of InlF, since a deletion mutant of *Lm* lacking InlF did not display reduced invasion rates.

## 1. Introduction

*Listeria monocytogenes* (*Lm*) is a ubiquitous, Gram-positive, facultative anaerobe bacterium primarily recognized as a food contaminant with a high tolerance for various environmental hazards (low temperature, low pH, and high salinity) [1]. It is also an opportunistic pathogen able to enter and survive within various phagocytic and non-phagocytic cells and a causative agent of listeriosis in humans and livestock [1]. Although the overall yearly number of globally reported cases of infection with *Lm* is relatively small, it is medically relevant due to a high incidence of blood-borne systemic spread of the bacterium in infected people, particularly those who are immunocompromised [1,2].

*Lm* is a food-borne pathogen, and food is primarily contaminated by *Lm* when it comes into contact with human or animal feces containing the bacteria, as the gastrointestinal tract is its most common habitat. The gastrointestinal tract is also the main point of entry into the host organism. In the majority of immunocompetent people, *Lm* does not spread beyond the gut after entering the body and is either asymptomatic or induces short-term gastroenteritis with subsequent clearance from the organism [1,3,4,5]. The pathogen is much more dangerous in immunocompromised people, especially those whose cell-mediated immune response is affected, since it is able to penetrate the gut barrier, invade, and survive within circulating phagocytes and traverse the vascular system to disseminate into various organs of the body, with the ones preferred initially being the liver and spleen [1,2,4,6]. Due to the inability of the immune system to fully clear the bacteria from the body within a reasonable amount of time, *Lm* might spread further into the central nervous system (CNS) and fetus (in pregnant women), which often results in long-term damage or even death [1,2,4].

Since the CNS is enveloped by specific barriers, to enter the brain from the blood, *Lm* must cross the barriers that border the bloodstream, the blood–brain barrier (BBB) and the blood–cerebrospinal fluid barrier (BCSFB) [4,7,8,9,10]. Whereas the BBB consists of the brain microvascular endothelium supported by astrocytes and pericytes, the BCSFB is built by the epithelium of the choroid plexus in the ventricles of the brain [11,12]. It has been shown that in vitro *Lm* can efficiently invade and pass through human brain microvascular endothelial cells (HBMEC), representing the BBB, and human choroid plexus epithelial papilloma (HIBCPP) cells, which are a substitute for the BCSFB [10,13,14,15].

*Lm* possesses a large number of virulence factors (VF) ranging from cell-type-specific adhesins to the cytolysin listeriolysin O (LLO) [1,2,16,17]. Internalins (Inl) form a protein family of listerial VF characterized by the presence of leucine-rich repeat (LRR) domains and Sec-dependent N-terminal signal peptides [16]. Important functions during host cell infection have been described for specific internalins. Internalin (InlA), the founding member of the internalin family, binds to E-cadherin (Ecad), a transmembrane protein normally found as a part of adherens junctions in barrier-forming epithelial cells [18]. In contrast, internalin B (InlB) interacts with the receptor tyrosine kinase Met [19]. From in vitro studies, only InlB is thought to mediate invasion into HBMEC, whereas both InlA and InlB are interdependently required for invasion of HIBCPP cells, which occurs in a polar fashion from the basolateral side [10,13,20,21].

InlF is comparable to InlA in both length and structure [16,22]. The only currently known interaction partner of InlF is vimentin, a type 3 intermediate filament protein found in mesenchymal cells, although the mechanism of this interaction is still unknown [23]. Data on the barrier-crossing properties of InlF are relatively scarce. The study by Ghosh et al. marks it as vital for breaching the BBB [23]. The results of their experiments, conducted both with the human brain microvascular endothelial cell line hCMEC/D3 and in mice, indicate that the interplay of InlF and vimentin is necessary for efficient invasion of the brain since the absence of either, or the obstruction of the contact between them, leads to severely hampered invasion into the cells or the brains of infected animals [23]. Data obtained by Bastounis et al. in human dermal microvascular endothelial cells (HMEC-1) confirm the importance of interaction between *Lm* and vimentin for endothelial cell invasion, although evidence for a role of InlF in this interaction was not reported. The authors pointed out that the possible significant differences between various human microvascular endothelial cell lines could affect which listerial VFs would be important for the invasion into them [24]. In contrast to the BBB, the role of the BCSFB during InlF-mediated CNS invasion has not yet been investigated.

We show that whereas HBMEC possess intracellular and surface vimentin (besides Met), HIBCPP cells do not express vimentin (but Met and Ecad). Accordingly, modulation of surface vimentin via treatment with WitA had no effect on HIBCPP cells but inhibited invasion of *Lm* into HBMEC. Invasion of both HBMEC and HIBCPP cells was independent of InlF. Therefore, invasion of *Lm* into HIBCPP cells is mainly mediated by InlA and InlB, and possibly further virulence factors, but does not involve an interplay between vimentin and InlF. In contrast, our data point to an important, but InlF-independent, role of vimentin during invasion of HBMEC.

## 2. Results

### 2.1. HBMEC and HIBCPP Cells Express Proteins Relevant for Lm Invasion

*Lm* utilizes proteins of the host cell for its own purposes, many of which are crucial for its successful colonization of and survival within the host [1]. Since the expression of these proteins is not uniform across all the cells of the body, it is necessary to ensure that the cell lines used as models for bacterial interaction, including adherence and invasion experiments, actually express them. Therefore, we analyzed the expression of proteins known to interact with the VFs of *Lm* investigated in this study (InlA, InlB, and InlF), which are Ecad, Met, and vimentin, respectively, in HBMEC and HIBCPP cells.

It has previously been shown that HIBCPP cells express Met and E-cadherin (Ecad) [10], and there is evidence that HBMEC express Met and vimentin [25,26]. Analysis via semiquantitative RT-PCR and western blotting (Figure 1) confirmed and demonstrated that while HIBCPP cells express E-cadherin (a marker of epithelial cells), they do not express vimentin (a marker of mesenchymal cells). On the other hand, HBMEC do not express E-cadherin, but do express vimentin (due to being cells of mesenchymal origin). Both cell lines express Met (which is normally expressed in both endothelial and epithelial cells), and both of them express ZO-1 (a tight junction marker protein), although the level of transcription and expression of the two is somewhat different between the two cell lines (Figure 1).

Immunostaining and visualization using immunofluorescence microscopy was further used to confirm the expression of Ecad, Met, and vimentin in HIBCPP cells and HBMEC, and to analyze their cellular localization. Whereas HIBCPP cells present the known basolateral localization of Ecad, no staining for Ecad can be detected in HBMEC (Figure 2). In contrast, Met can be visualized in both HIBCPP cells and HBMEC (Figure 3). Vimentin staining in HBMEC and HIBCPP cells confirms the presence of vimentin in HBMEC and its complete absence in HIBCPP cells (Figure 4). Analysis of samples which were permeabilized before antibody staining and of samples without permeabilization shows that vimentin is localized both intracellularly and on the cellular surface of HBMEC (Figure 4). Both HBMEC and HIBCPP cells display an expression of ZO-1 that is indicative of the presence of tight junction strands (Figure 2, Figure 3 and Figure 4).

### 2.2. Modulation of Surface Vimentin Reduces the Invasion of Lm into HBMEC

Vimentin was implicated as an important target for *Lm* (strain 10403S) during the invasion of microvascular endothelial cells, including BBB models, in vitro (murine bEnd.3, human hCMEC, and HMEC-1 microvascular endothelial cell lines) and in vivo (murine infection model) and is also a known interaction partner for a number of bacterial pathogens [23,24,27]. To determine its relevance for invasion of *Lm* into HBMEC and HIBCPP cells, the cells were infected with wild-type *Lm* EGD-e after pre-incubation with increasing concentrations of withaferin A (WitA), a known surface vimentin modulator [23,24].

It has previously been shown that *Lm* can invade HBMEC from the apical side [13], whereas invasion of HIBCPP cells by *Lm* occurs in a polar fashion from the basolateral side due to the basolateral localization of Ecad and Met [10]. Therefore, apical infection experiments with HBMEC were performed with cells grown in well cultures, whereas basolateral infection of HIBCPP cells was achieved in an inverted cell culture filter insert model, as previously described [10,28,29]. HIBCPP cells grown on cell culture filter inserts develop a strong barrier function, displaying a high transepithelial electrical resistance (TEER) and low permeability for FITC-labelled inulin. Following pre-incubation with WitA (1 µM, 5 µM or 10 µM) for 0.5 h, cells were infected with *Lm* EGD-e over a time period of 4 h. Subsequently, invasion rates were determined using double immunofluorescence microscopy. Analysis of barrier function of HIBCPP cells showed that average TEER values stayed above 300 Ω × cm^2^, and the average Inulin-FITC flux was below 4% except for HIBCPP cells infected and treated with 10 µM WitA, indicating some impact on barrier function under this condition (Appendix A).

The results obtained during the infection experiments show a statistically significant WitA-concentration-dependent decrease in invaded *Lm* in HBMEC (Figure 5A). In contrast, treatment of HIBCPP cells with WitA resulted in no statistically significant decrease in invasion with all WitA concentrations applied (Figure 5B). Analysis via live/dead assay showed that co-incubation with WitA in presence of *Lm* does not affect the survival of HBMEC or HIBCPP cells (Appendix A).

### 2.3. Different Internalins Are Required for Invasion of Lm into HBMEC and HIBCPP Cells

It was previously reported that listerial invasion into HBMEC requires InlB but not InlA and that *Lm* uses both InlA and InlB interdependently for efficient invasion into HIBCPP cells [10,13]. We were now interested in evaluating the requirement of InlF during infection of HBMEC and HIBCPP in comparison to InlA and InlB. For this purpose, HBMEC grown in well cultures and HIBCPP cells cultivated in the inverted cell culture filter insert model were infected with the wild-type bacteria of the EGD-e strain (*Lm* EGD-e wt) as well as with deletion mutants for internalins A, B, and F (*Lm* EGD-e Δ*inlA*, *Lm* EGD-e Δ*inlB*, *Lm* EGD-e Δ*inlAB,* and *Lm* EGD-e Δ*inlF*). Analysis of barrier function of HIBCPP cells showed that average TEER values remained at or around 200 Ω × cm^2^, and the average Inulin-FITC flux was around or below 4% (Appendix A). The more recognizable impact on barrier function by *Lm* EGD-e wt and *Lm* EGD-e *ΔinlF* is probably due to the higher invasion of these strains.

The results confirm that *Lm* depends strongly on InlB for invasion into the HBMEC (Figure 6A) and relies interdependently on both InlA and InlB for invasion into the HIBCPP cells (Figure 6B). Unexpectedly, deletion of InlA also caused reduced invasion into HBMEC, although less pronounced than the deletion of InlB (Figure 6A). Deletion of InlF did not cause a change in infection rates of either HBMEC or HIBCPP cells (Figure 6).

## 3. Discussion

The BBB and the BCSFB are potential entry gates for *Lm* into the human CNS. Invasion of cells of either the BBB or the BCSFB presents a putative first step during brain entry by *Lm*, which could subsequently continue passing into the CNS. Whereas the microvascular endothelial cells of the brain are the main contributors to the BBB, barrier function at the BCSFB is executed by the choroid plexus epithelium [11,12]. In this study we used HBMEC and HIBCPP cells as established in vitro models of the BBB and the BCSFB, respectively [29,30,31]. An arsenal of VF has been described to be involved during host cell invasion by *Lm* [17]. Among these, the Inl protein family is of special interest since its members InlA, InlB, and InlF are implicated in the process of CNS entry [2,16]. InlA, InlB, and InlF have been shown to interact with Ecad, Met, and vimentin, respectively, on the surface of host receptors [18,19,23].

When investigating the roles of these Inls during host cell invasion, it is important to analyze whether the employed model systems express their respective interaction partners. Here, we show that HBMEC express Met and vimentin, but not Ecad (Figure 1, Figure 2 and Figure 3). This confirms previous evidence for the expression of Met and vimentin in HBMEC [25,26]. In contrast, HIBCPP cells expressed Ecad and Met as shown previously [10], but vimentin was not detected (Figure 1, Figure 2 and Figure 3). There are conflicting results concerning the expression of vimentin in the choroid plexus epithelium [32,33,34]. Primary mouse choroid plexus epithelial cells express Ecad, ZO-1 and cytokeratin, but not vimentin, and display a barrier function [35]. In contrast, following immortalization expression of cytokeratin and Ecad is lost. Instead, expression of the mesenchymal marker proteins vimentin and N-cadherin is observed. Additionally, a mouse choroid plexus carcinoma cell line lacked expression of cytokeratin and Ecad and failed to establish junctional complexes, but was positive for vimentin and did not develop a barrier function [35]. We would like to point out that HIBCPP cells resemble the primary mouse choroid plexus epithelial cells since they express Ecad and ZO-1, but not vimentin, and develop a strong barrier function.

The intermediate filament protein vimentin is found primarily in mesenchymal (e.g., endothelial) cells and can be located both intracellularly and extracellularly. It serves multiple roles inside the cell, including but not limited to the provision of flexibility to the cytoskeleton, assistance in the anchoring of cellular organelles, and lipid droplet formation [36,37]. Extracellular vimentin is of special interest during pathogenic host cell invasion since it was found as an interaction partner for several intracellular pathogens—dengue virus (via NS4a), *E. coli* K1 (via IbeA), group B *Streptococcus* (via BspC), and *Lm* (via InlF or independently of InlF) [23,24,38,39,40]. The expression of vimentin by HBMEC, including the detection of surface-located vimentin (Figure 4), raised the question of to which extent vimentin is involved during the invasion of *Lm* into HBMEC. Treatment of HBMEC with increasing concentrations of withaferin A (WitA), a known surface vimentin modulator, caused a strong decrease in listerial invasion (Figure 5). Similar findings have been described in human microvascular endothelial cells (HMEC-1) by Bastounis et al. [24], and a dose-dependent decrease in *Lm* invasion in different cell lines by WitA treatment was also demonstrated by Ghosh et al. [23]. It has also been published that treatment of HBMEC with WitA caused an increasing reduction of vimentin levels and blocked invasion by *E. coli* K1 via IbeA [38].

Analysis of different Inl deletion mutants confirmed previous findings that the deletion of InlA, InlB or both causes a similar decrease in bacterial invasion into HIBCPP cells, implying an interdependent mode of action for InlA and InlB [10]. Invasion into HBMEC was also strongly inhibited by deletion of InlB (Figure 6), again confirming previous findings [13]. Compared to the wildtype, we found a partial decrease in invasion of HBMEC by a ΔInlA mutant. This result differs from the data of Greiffenberg et al., who found no impact of InlA deletion on invasion into HBMEC [13]. Notably, Parida et al. described a partial inhibition of *Lm* invasion into human umbilical vein endothelial cells (HUVECs) by deletion of InlA [41]. We speculate that the observed variances are due to differences in experimental setups regarding the exact nature of HBMEC and bacterial strains employed. When analyzing the impact of InlF deletion, we found that InlF was not required for invasion of *Lm* into both HBMEC and HIBCPP cells (Figure 6). We hypothesize that the remaining capacity of *Lm* for invasion into HIBCPP cells in absence of InlA and InlB is mediated by listerial VFs other than InlF, such as further Inls (InlC, InlGHE or others) or LLO.

Whereas the role of InlF during infection of choroid plexus epithelial cells has not been investigated before, InlF was implicated as a key listerial factor in interaction with vimentin during invasion of *Lm* into the brain microvascular endothelial cell line hCMEC/D3 [23]. Interestingly, these experiments were performed under conditions of inhibition of Rho-associated protein kinases (ROCKs) [23], which are involved in actin cytoskeleton rearrangements [42], since previous data had shown that inhibition of ROCK activity results in cell-type-specific increased host cell binding and entry by *Lm* [43]. For example, ROCK inhibition led to increased binding and invasion in murine fibroblast (L2) and hepatocyte (TIB 75) cell lines, which was dependent on InlF. ROCK inhibition also increased binding and entry in human fibroblast (WI38) and only binding in epithelial (HeLa) cell lines, but this effect was independent from InlF [43]. In a recent publication, Ling et al. found that InlF contributes to adhesion and invasion of macrophages but was not involved in adherence or invasion during infection of five non-phagocytic cell lines. However, a function of ROCK was not analyzed [44]. Similarly, we observe InlF independent invasion of *Lm* into HBMEC and HIBCPP cells, but determination of a possible role of ROCKs would require further investigation.

The data of Ghosh et al. pointed to the roles of InlF and vimentin during listerial colonization of the brain [23]. In contrast, the vimentin-mediated infection of HMEC-1 cells described by Bastounis et al. was independent of InlF [24]. The authors noted that possible significant, but as-of-yet unidentified, differences between various human microvascular endothelial cell lines could affect which listerial VFs are important for invasion. Since a decreased amount of surface vimentin caused decreased adhesion of both *Lm* and the non-pathogenic relative *Listeria inoccua*, a potential interaction partner could be expressed by both bacterial species. Furthermore, a role of cell surface vimentin as a host cell receptor mediating extracellular matrix stiffness-dependent adhesion of *Lm* was suggested [24]. Since invasion of *Lm* into HBMEC was sensitive to WitA but independent of InlF, we speculate that this process might involve a role of vimentin related to regulation of extracellular matrix stiffness.

In summary, we show that HBMEC and HIBCPP cells differentially express target proteins for listerial virulence factors, with vimentin only present on HBMEC. Deletion of InlF shows that *Lm* does not require InlF for invasion of HBMEC or HIBCPP cells. Since HIBCPP cells do not express vimentin, InlF- and vimentin-independent invasion of *Lm* is mainly mediated by InlA and InlB and possibly further virulence factors. Entry into HBMEC requires an InlF-independent role of vimentin, which might involve the previously described regulation of extracellular matrix stiffness. This process could require specific virulence factors of *Lm*, possibly binding to vimentin, further underlining the differences of listerial invasion at the BBB and BCSFB in vitro.

## 4. Materials and Methods

### 4.1. Bacterial Strains, Construction of EGD-e ΔinlF, and Bacterial Growth Conditions

*Lm* serotype 1/2a strain EGD-e and its isogenic deletion mutants EGD-e Δ*inlA*, EGD-e Δ*inlB* and EGD-e Δ*inlAB* have been described before [41,45,46].

The isogenic strain EGD-e Δ*inlF* was constructed by generating the 5′ (using primers P1 and P2) and the 3′ (using primers P3 and P4) flanking regions of *inlF* (lmo0409). The primers used to generate the flanking regions are shown in Table 1. The PCR fragments were purified and ligated into the temperature-sensitive suicide vector pAUL-A. The presence of the insert was screened by using the primer pair M13 Forward/M100 and M13 Reverse/M101 (Table 1). The annealing temperature of the PCR reactions was 55 °C. Subsequently, *E. coli* InvαF′ electrocompetent cells were transformed with the plasmid containing the correct insert. Plasmid DNA of pAUL-A bearing the fragments was isolated from the recombinant *E. coli* cells and used to transform *L. monocytogenes* to generate the chromosomal Δ*inlF* deletion mutant as described previously [47]. The deletion of *inlF* was confirmed by sequencing the PCR products by using the flanking primers P7 and P8.

All strains were stored in brain heart infusion (BHI) medium (BD, Franklin Lakes, NJ, USA) containing 30% glycerol. In preparation for infection experiments bacteria were incubated in BHI medium for 6 h at 37 °C under slight agitation to mid-logarithmic phase. Subsequently, bacteria were washed two times in serum-free medium (SFM) and adjusted to an optical density at 600 nm (OD_600_) values corresponding to a concentration of 1 × 10^8^ CFU/mL (EGD-e wt, OD_600nm_ of 0.2; EGD-e Δ*inlA*, OD_600nm_ of 0.4; EGD-e Δ*inlB*, OD_600nm_ of 0.1; EGD-e Δ*inlAB*, OD_600nm_ of 0.6; EGD-e Δ*inlF*, OD_600nm_ of 0.2).

### 4.2. Cell Culture of HBMEC and HIBCPP Cells

HIBCPP cells, a human choroid plexus papilloma epithelial cell line, present an established in vitro model of the BCSFB [29,31]. HBMEC are an immortalized human brain microvascular endothelial cell line established as an in vitro model of the BBB [30]. Cell medium for growth of HIBCPP cells was DMEM/F12 medium supplemented with 4 mM L-glutamine, 15 mM HEPES, human recombinant insulin solution (10 mg/mL) and 5 mg mL^−1^ insulin, penicillin (100 U mL^−1^), and streptomycin (100 mg mL^−1^) (HIBCPP cell-medium) containing 10% heat-inactivated fetal calf serum (FCS). Cell medium for growth of HBMEC was DMEM/F-12 medium supplemented with 4 mM L-glutamine, 15 mM HEPES and 5 mg mL^−1^ insulin, penicillin (100 U mL^−1^), and streptomycin (100 mg mL^−1^) (HBMEC medium) containing 10% heat inactivated FCS. During infection experiments media contained only 1% FCS.

For infection experiments HIBCPP cells were grown in the inverted cell culture insert model as previously described [28,29].

### 4.3. Measurement of Barrier Function

TEER measurements of HIBCPP cell layers grown on cell culture inserts was performed at the beginning and the end of experiments with a STX 01 chopstick voltohmmeter electrode connected to a Millicell^®^-ERS voltohmmeter (Millipore, Schwalbach, Germany). Measurement of the permeability of HIBCPP cell layers was performed following infection experiments to evaluate the status of the cellular barriers of cells on individual cell culture insert filters. Before the infection of the cells at the start of the experiment FITC-labelled inulin (average molecular weight of 3000 to 6000; Sigma, Deisenhofen, Germany) was added into the upper compartment of the cell culture filter inserts. At the end of the experiment, the passage from the upper compartment to the lower compartment was monitored with a Tecan Infinite M200 Multiwell reader (Tecan, Switzerland) as described previously [10]. Barriers with a permeability below 4% during the course of the experiment were considered stable.

### 4.4. Semiquantitative RT-PCR

For RNA preparation, cells were initially washed with PBS. Subsequently, RNeasy Micro or RNeasy Mini Kits (depending on the available starting material across experiments) were used in accordance with the manufacturer’s (Qiagen, Hilden, Germany) instructions for RNA isolation. Following the purification, the concentration and quality of purified RNA were determined with a NanoDrop^®^ ND1000 spectrophotometer (Peqlab Biotechnology, Erlangen, Germany). For the generation of cDNA, the AffinityScript QPCR cDNA Synthesis Kit (Agilent Technologies, Santa Clara, CA, USA) was used with 500 ng of total RNA in accordance with the manufacturer’s instructions. The obtained cDNA was used as a template for semi-quantitative PCR with the Taq DNA Polymerase Kit (Qiagen, Hilden, Germany). To enable relative assessment of the initial quantity of the gene material in the sample, PCR product was analyzed via gel electrophoresis and ethidium bromide staining after 26 and 30 PCR cycles, respectively. The following PCR conditions were used: initial denaturation (94 °C, 2 min) and subsequently 30 cycles of denaturation (94 °C, 30 s), annealing (58–62 °C, depending on the primer pair, 30 s), extension (72 °C, 1 min). Subsequently, a final extension (72 °C, 7 min) was performed. The following PCR primers were used: vimentin, AGAGAGAGGAAGCCGAAAAC (forward), TGGATTTCCTCTTCGTGGAGTT (reverse); E-cad, CCTGCCAATCCCGATGA (forward), TGCCCCATTCGTTCAAGTA (reverse); Met ATCTTGGGACATCAGAGGGT (forward), TCGTGATCTTCTTCCCAGTGA (reverse); ZO-1, GCCAAGCAATGGCAGTCTC (forward), CTGGGCCGAAGAAATCCCATC (reverse).

### 4.5. Immunoblot

To obtain protein extracts for Western blotting, cells were washed with PBS twice and lysed in RIPA buffer after the removal of all excess PBS. Proteins of 10 to 20 µg of protein extract were separated on 4–12% Bis-Tris (Invitrogen, Karlsruhe, Germany) and subsequently transferred onto nitrocellulose membranes via electroblotting. Membranes were incubated for 1 h in blocking solution (5% milk powder solution in dH_2_O), and proteins were detected with the following antibodies: chicken anti-vimentin (BioLegend, San Diego, CA, USA), goat anti-Met (Abcam, Cambridge, UK), mouse anti-Ecad (BD, Franklin Lakes, NJ, USA), and rabbit anti-ZO-1 (Invitrogen, Carlsbad, CA, USA). The Immobilon Western Kit (Millipore, Schwalbach, Germany) was used for the visualization of detected proteins.

### 4.6. Immunofluorescence

Immunofluorescence analysis of HBMEC grown in chamber slides and HIBCPP cells cultivated in the inverted cell culture insert model was performed as previously described [29]. The following antibodies were used: primary antibodies: chicken anti-vimentin (BioLegend, San Diego, CA, USA), goat anti-Met (Abcam, Cambridge, UK), mouse anti-Ecad (BD, Franklin Lakes, NJ, USA), and rabbit anti-ZO-1 (Invitrogen, Carlsbad, CA, USA); secondary antibodies: goat anti-chicken Alexa Fluor^®^ 594, donkey anti-goat Alexa Fluor^®^ 594, goat anti-mouse Alexa Fluor^®^ 594, chicken anti-rabbit Alexa Fluor^®^ 488, and donkey anti-rabbit Alexa Fluor^®^ 488 (all Invitrogen, Carlsbad, CA, USA). Nuclei were stained with 4′-6-diamidino-2-phenylindole dihydrochloride (DAPI) (1:50,000 in PBS/1% BSA) (Merck, Darmstadt, Germany). Densitometric analysis of Western blot bands normalized to actin was performed using the ImageJ software 1.53e [48].

### 4.7. Infection of HBMEC and HIBCPP Cells with Lm

During the infection experiments, bacteria were co-incubated with the cells in either 1% HIBCPP medium or 1% HBMEC medium, depending on the cell line used. HBMEC at a confluency and HIBCPP cells grown in the inverted cell culture insert model with a TEER between 270 and 800 Ω × cm^2^ were infected with the different *Lm* strains at a multiplicity of infection (MOI) of 100 for 4 h. When infecting HIBCPP cells, bacteria were added into the upper compartment of the filter system to allow infection from the basolateral cell side as previously described [28,29]. In some experiments, WitA (Merck, Darmstadt, Germany) was added to the cells in various concentrations 30 min before addition of the bacteria.

### 4.8. Measurement of Cell Viability

A LIVE/DEAD Viability/Cytotoxicity Kit for mammalian cells (Invitrogen, Karlsruhe, Germany) was used in experiments applying WitA and bacterial infection to track the viability of HBMEC and HIBCPP cells throughout the experimental time period. The kit was used according to the manufacturer’s instructions. For documentation of the results, fluorescence microscopic pictures were taken.

### 4.9. Determination of Bacterial Invasion by Double Immunofluorescence

Determination of bacterial invasion rates in HBMEC grown in chamber slides and HIBCPP cells cultivated in the inverted cell culture insert model was performed as previously described for HIBCPP cells [10,29]. The following antibodies were used: primary antibody: rabbit anti-*Listeria monocytogenes* (Meridian Life Sciences, Memphis, TN, USA); secondary antibodies: donkey anti-rabbit Alexa Fluor^®^ 488 and chicken anti-rabbit Alexa Fluor^®^ 594 (both Invitrogen, Carlsbad, CA, USA) for staining of intra- and extracellular bacteria, respectively. The actin cytoskeleton was stained with Alexa Fluor™ 660 Phalloidin (1:250) (Invitrogen, Carlsbad, CA, USA), nuclei were stained with 4′-6-diamidino-2-phenylindole dihydrochloride (DAPI) (1:50,000 in PBS/1% BSA) (Merck, Darmstadt, Germany).

### 4.10. Statistical Analysis

Statistical analysis of the data presented in this study was performed with SAS, release 9.4 (SAS Institute Inc., Cary, NC, USA). For results presented in Figure 5 and Figure 6, a one-way ANOVA was performed to compare mean values of multiple differently treated groups. In case of a significant result, Dunnett’s post hoc test was used in order to compare each treated group with the control group or the wild type, respectively.

*p*-values were considered as being significant (*), very significant (**), or extremely significant (***/****) when <0.05, <0.01, or <0.001/0.0001, respectively. *p*-values > 0.05 were considered not significant (ns).

## Figures and Tables

**Figure 1 ijms-23-12908-f001:**
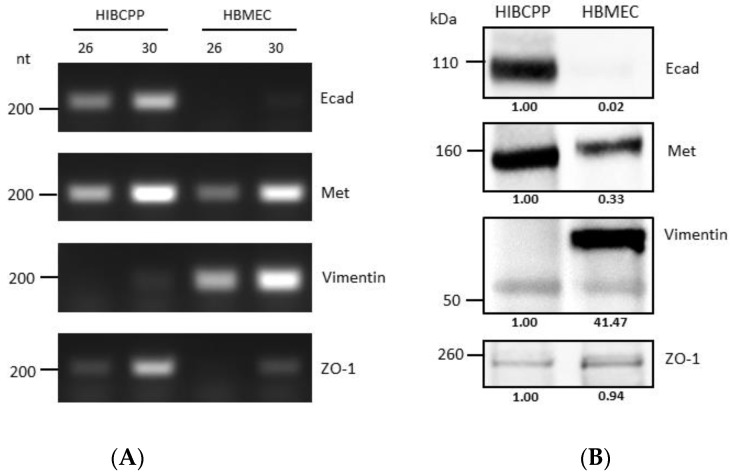
Differences in molecular expression patterns of VF target proteins between HIBCPP cells and HBMEC. (**A**) Transcription levels of genes of interest in HIBCPP cells and HBMEC, as obtained through semi-quantitative PCR (numbers 26 and 30 denote the number of PCR cycles). Gene names are indicated at the right side of the panels (**B**) Analysis of the protein expression levels of proteins of interest in HIBCPP cells and HBMEC via western blotting. Gene names are indicated at the right side of the panels. The presented data were selected as representative of the average outcome of multiple performed experiments.

**Figure 2 ijms-23-12908-f002:**
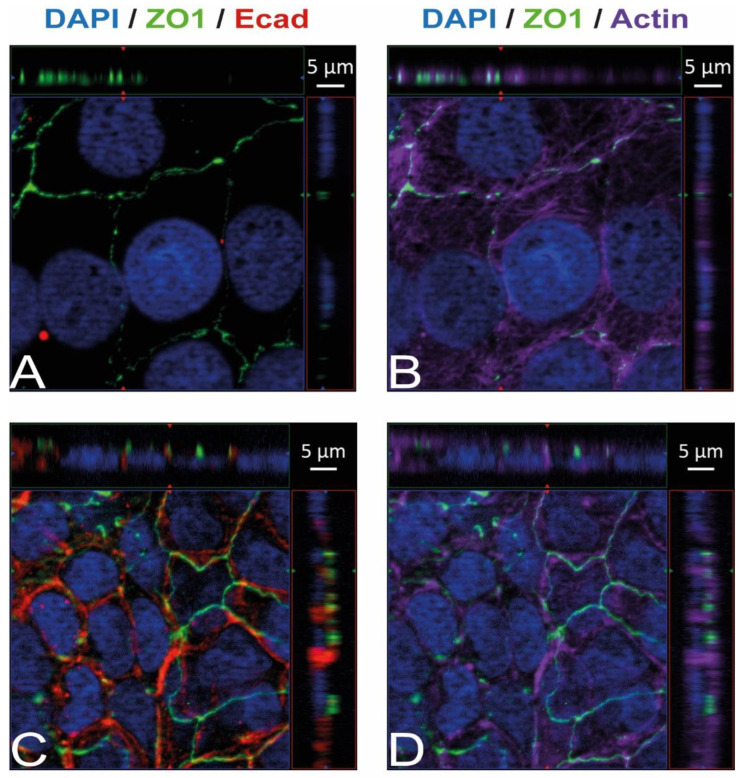
Ecad is present in HIBCPP cells but absent in HBMEC. Immunofluorescence staining was performed with HBMEC (**A**,**B**) and HIBCPP cells (**C**,**D**). Ecad staining (**A**,**C**) is shown in red; the phalloidin-stained actin cytoskeleton (**B**,**D**) is shown in purple. Tight junctions are visualized via the staining of ZO1 (green); nuclei are stained with DAPI (blue). All panels represent Apotome images. The center part of each image displays an xy enface view presented as a maximum intensity through the *z*-axis of selected slices. The top and right-side parts of each panel are cross sections through the z-plane of multiple optical slices. The apical sides of HBMEC and HIBCPP cells are oriented toward the top of the top part and towards the right of the right part of each panel, respectively.

**Figure 3 ijms-23-12908-f003:**
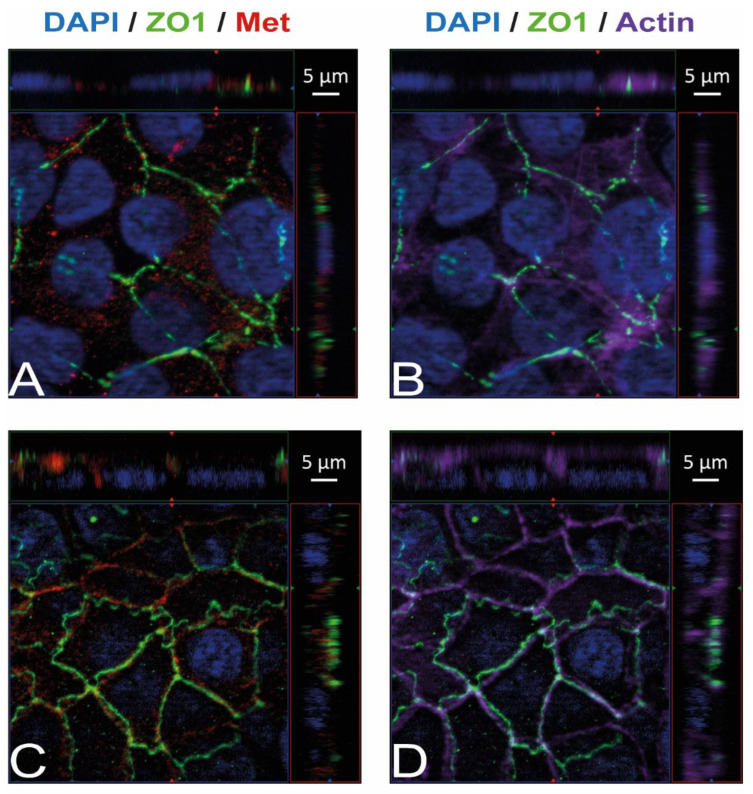
Met is present in HBMEC and HIBCPP cells. Immunofluorescence staining was performed with HBMEC (**A**,**B**) and HIBCPP cells (**C**,**D**). Met staining (**A**,**C**) is shown in red; the phalloidin-stained actin cytoskeleton (**B**,**D**) is shown in purple. Tight junctions are visualized via the staining of ZO1 (green); nuclei are stained with DAPI (blue). All panels represent Apotome images. The center part of each image displays an xy enface view presented as a maximum intensity through the *z*-axis of selected slices. The top and right-side parts of each panel are cross sections through the z-plane of multiple optical slices. The apical sides of HBMEC and HIBCPP cells are oriented toward the top of the top part and towards the right of the right part of each panel, respectively.

**Figure 4 ijms-23-12908-f004:**
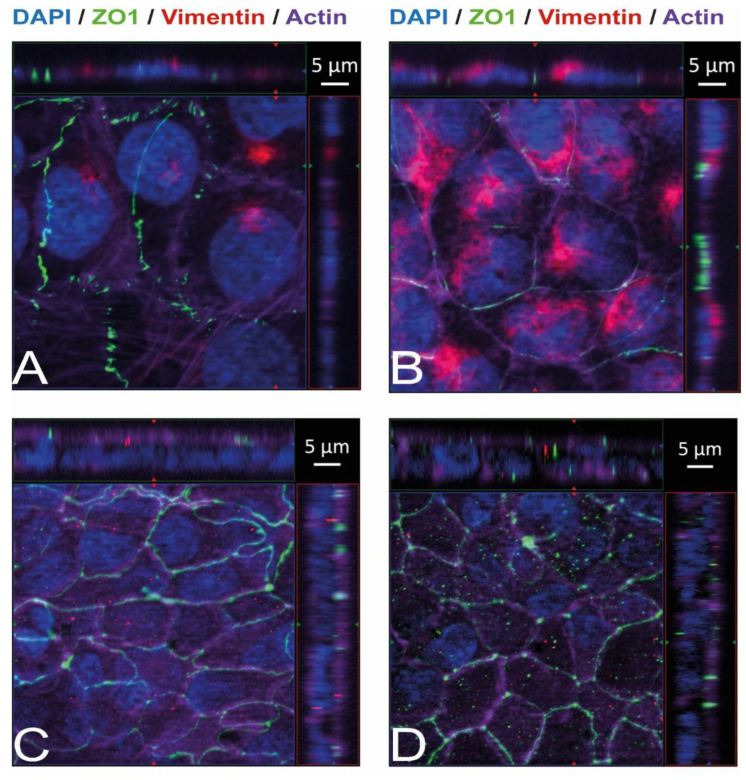
Vimentin, both intracellularly and on the cellular surface, is present in HBMEC but absent in HIBCPP cells. Immunofluorescence staining was performed with HBMEC (**A**,**B**) and HIBCPP cells (**C**,**D**). Vimentin staining is shown in red, the phalloidin-stained actin cytoskeleton is shown in purple, tight junctions are visualized via the staining of ZO1 (green), and nuclei are stained with DAPI (blue). The center part of each image displays an xy enface view presented as a maximum intensity through the *z*-axis of selected slices. The top and right-side parts of each panel are cross sections through the z-plane of multiple optical slices. The apical sides of HBMEC and HIBCPP cells are oriented toward the top of the top part and towards the right of the right part of each panel, respectively. Samples presented on the left (**A**,**C**) were not permeabilized before the binding of the anti-vimentin antibody and, therefore, show only vimentin present on the surface of the cells. Samples presented on the right (**B**,**D**) were permeabilized before the binding of the anti-vimentin antibody and, therefore, show the total vimentin present in the cells.

**Figure 5 ijms-23-12908-f005:**
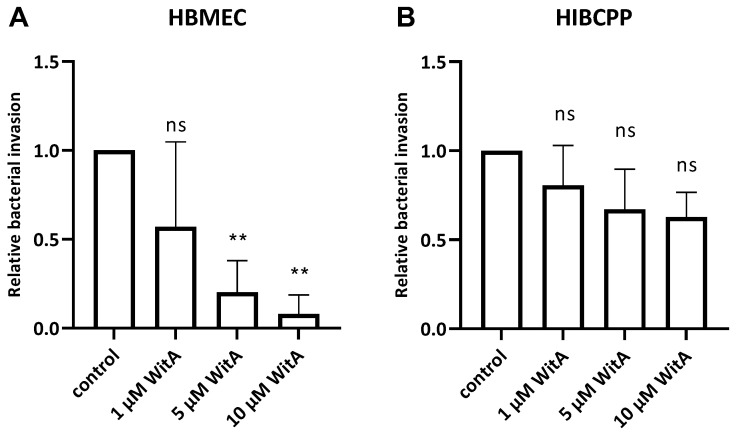
*Lm* requires surface vimentin for entry into HBMEC. HBMEC grown in chamber slides (**A**) and HIBCPP cells grown in the inverted cell culture filter system (**B**) were pre-incubated with WitA (/, 1 µM, 5 µM or 10 µM) for 0.5 h and subsequently infected with wild-type *Lm* EGD-e (MOI = 10) and co-incubated for 4 h. Invasion rates were determined using double immunofluorescence microscopy. Shown is the relative bacterial invasion normalized to the untreated control. The results of at least three independent experiments performed at least in duplicate are shown. ** = very significant (*p* < 0.01), ns = not significant (significance determined in relation to the invasion of the wild-type bacteria).

**Figure 6 ijms-23-12908-f006:**
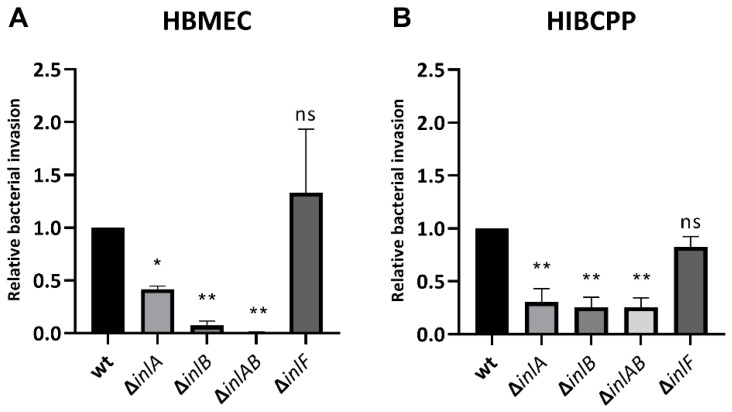
InlF is not required for invasion into HBMEC or HIBCPP cells. HBMEC grown in chamber slides (**A**) and HIBCPP cells grown in the inverted cell culture filter system (**B**) were then infected with wild type and the indicated mutant *Lm* EGD-e (MOI = 10) for 4 h. Invasion rates were determined via double immunofluorescence microscopy. Shown is the relative bacterial invasion normalized to the wild type. The results of at least 2 independent experiments performed in duplicate (HBMEC) or triplicate (HIBCPP cells) are shown. * = significant (*p* < 0.05), ** = very significant (*p* < 0.01), ns = not significant (significance determined in relation to the invasion of the wild-type bacteria).

**Table 1 ijms-23-12908-t001:** Primers used for generation of EGD-e Δ*inlF*.

Primer	5′-3′ Sequence
P1-f	TCGTAGAGATAAAATCGACAAACAA
P2-r	TTTTTAATTA_ATTAGTCTTTCCTTTCATTA
P3-f	AAAGACTAAT_TAATTAAAAAACCCAGCATT
P4-r	TCATCTGGGACAGTTGAAGG
P7-f	TCCCGCTAACTGGTCATAAAGGC
P8-r	ACTGCGGGAAGTTGTGCGTAC
M13 Forward/M100	GTAAAACGACGGCCAG
M13 Reverse/M101	CAGGAAACAGCTATGAC

## Data Availability

Data sharing is not applicable to this article.

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
