# Peer review of "Different Involvement of Vimentin during Invasion by *Listeria monocytogenes* at the Blood–Brain and the Blood–Cerebrospinal Fluid Barriers In Vitro"

_ijms, 2022, doi:10.3390/ijms232112908_

Round 1

Reviewer 1 Report (Previous Reviewer 2)

The authors have responded to all suggestions/queries. I endorse the manuscript for publication. 

Thank you

Author Response

The authors again want to express their gratidute for the helpful comments and cooperation. 

Reviewer 2 Report (Previous Reviewer 1)

The authors have focused on the study Invasion to cells of either the BBB or the BCSFB is a potential first step during CNS entry by the Gram-positive bacterium Listeria monocytogenes. Based on the fact that the bacterium possesses several virulence factors mediating host cell entry.

The authors have looked to investigate the roles of  family member is internalin and vimentin during CNS invasion by Listeria monocytogenes, which targets the intermediate filament protein vimentin. The experimental base used in the study was adequate for the study carried out. Possibly the nesting temperatures of the primers could be reflected to create those mutants with which the study worked in addition to referencing.

Reflect that the results and discussion are consistent.

The introduction is adequate for the study carried out and the conclusions contain what is relevant to the study.

Author Response

According to the suggestion we have added the temperatures of the primers in the m&m section on page 12.

The authors again want to thank the reviewer for the effort and helpful comments. The manuscript was decisively improved by the cooperation. 

This manuscript is a resubmission of an earlier submission. The following is a list of the peer review reports and author responses from that submission.

Round 1

Reviewer 1 Report

The article reflects the difference in the expression of target proteins for listerial virulence factors. With vimentin only present in human brain microvascular endothelial cells during invasion by Listeria monocytogenes

The introduction contains the information and bibliography necessary to focus on the topic of the article. Regarding the results, they clearly reflect the findings supported by graphs and images. In this section, comment that the authors could reflect the size of the genes shown and of the transcribed protein (figure 1). Likewise, in figures 5 and 6 the statistics of the experiments are reflected as "ns = not significant" but the length of the line is large if compared to n "*, **". Perhaps it could be explained in text why it was considered "ns"

Regarding the discussion section, despite reflecting the results of other investigations, I missed the reference to the graphs/images shown in the results.

And finally, in the methodology section, in the subsection "Bacterial strains and growth conditions" in line 333, perhaps despite referencing it, it could be commented if there is a difference in the culture or at least growth temperature... well, when reading the The title of the subsection does not correspond to the information given to the reader... or the title could be modified by referring to the construction of the strains

Reviewer 2 Report

The study entitled “Different involvement of vimentin during invasion by Listeria monocytogenes at the blood-brain and the blood-cerebrospinal fluid barrier in vitro” by Franzo and colleagues reported the important role of vimentin in Lm infection. The study is good, and well written. However, there are some suggestions/queries if authors answers, that would be great.

  1. In Figure 1, there seems a very significant difference in expression of these proteins. Please provide the replicates of western blot and quantification data for the blots.
  2. Please provide the permeability data (with absolute TEER value and FITC-inulin) for all the experiments.
  3. Did the authors note any significant difference in adhesion of Lm on HBMEC and HIBCCP cells with or without WitA treatment?
  4. Does WitA changes the expression/activity of Met, vimentin and E cadh proteins at the concentration and time used for treatment? If there is any literature available for that, please include in discussion.
  5. Does WitA affects the viability of Lm? (Please provide the data for all mutants at tested concentration of Wit A with similar incubation time that was used for invasion experiments.
  6. Figure 1-4 can be combined and Figure 5-6 can be combined together.